# Factors Affecting the Flight Capacity of Two Woodwasp Species, *Sirex noctilio* F. (Hymenoptera: Siricidae) and *Sirex nitobei* M. (Hymenoptera: Siricidae)

**DOI:** 10.3390/insects14030236

**Published:** 2023-02-27

**Authors:** Xiaobo Liu, Juan Shi, Lili Ren, Youqing Luo

**Affiliations:** 1Beijing Key Laboratory for Forest Pest Control, College of Forestry, Beijing Forestry University, Beijing 100083, China; 2IFOPE, Sino-French Joint Laboratory for Invasive Forest Pests in Eurasia, Beijing Forestry University—French National Research Institute for Agriculture, Food and Environment (INRAE), Beijing 100083, China

**Keywords:** *Sirex noctilio*, *Sirex nitobei*, flight capacity, flight mills, *Deladenus siricidicola*

## Abstract

**Simple Summary:**

*Sirex noctilio*, an invasive woodwasp, together with *Sirex nitobei*, a native woodwasp, are potentially harmful to numerous pines in China. To investigate the flight capacity of these two woodwasp species in China, we performed laboratory measurements with a tethered-flight mill system. Our results indicate that *S. noctilio* can fly over a great distance and for a longer duration than *S. nitobei*. Nematode parasitism had no significant effect on the flight capacity of *S. noctilio* and *S. nitobei*. For both sexes of *S. noctilio*, post-eclosion-day (PED) age and body mass significantly influenced their flight capacity, and their flight capacity decreased with PED age. For *S. nitobei*, PED age did not significantly affect its flight capacity, while body mass significantly affected its flight capacity. From multiple regression analysis, PED age and body mass were major factors influencing the flight capacity of both *Sirex* species.

**Abstract:**

*Sirex noctilio* F. is an invasive woodwasp that causes pine mortality in plantations in China. *Sirex nitobei* M. is a native woodwasp in large areas of China. In this study, the flight capacity of the two woodwasps was studied and compared using a tethered-flight mill system to find individual factors affecting the flight capacity. After flight bioassays, woodwasps were dissected to determine nematode infestation. Post-eclosion-day (PED) age significantly influenced the flight capacity of *S. noctilio* females and males; as woodwasps become older, their flight capacity decreased. For *S. nitobei*, PED age did not significantly affect their flight capacity. In general, the flight capacity of *S. noctilio* was greater than that of *S. nitobei*. Females flew further and for longer than males for both *Sirex* species. The *Deladenus* spp. parasitism status of the two *Sirex* species did not significantly affect their flight performance parameters. PED age and body mass were key individual factors significantly affecting the flight capacity of the two *Sirex* species. In this study, detailed and accurate tethered-flight parameters of *S. noctilio* and *S. nitobei* were obtained. Although this is different from natural flight, it also provides us substantial laboratory data on their flight capacity, and facilitates risk analysis of the two woodwasp species.

## 1. Introduction

*Sirex noctilio* Fabricius (Hymenoptera: Siricidae) and *Sirex nitobei* Matsumura (Hymenoptera: Siricidae) are two major woodwasp pests in northeast China. *S. noctilio* originated in Eurasia and North Africa and is currently one of the most important threats to pine forests all over the globe. It has been introduced into several countries around the world, first invading New Zealand in 1900, Australia in 1952, Uruguay in 1980, Argentina in 1985, Brazil in 1988, South Africa in 1995, Chile in 2000, the United States in 2005, and China in 2013 [1,2,3,4,5,6]. In the northeast of China, *S. noctilio* generally produces one generation per year. The adult emergence period lasts from late July to early September, and the emergence peak occurs from the beginning of August to the end of August. The spread of non-native species in short ranges is largely determined by the flight capacity of the species [7]. Several researchers have explored the effects of individual size, body weight, temperature, photoperiod, mating, and nematode infestation on the flight capacity of *S. noctilio* [8,9,10,11]. However, the impact of post-eclosion-day (PED) age on the flight capacity of *S. noctilio* remains unknown. Studies have shown high variability in flight ability between *S. noctilio* individuals, which is considered strongly related to their body size [9,10,11,12,13]. It is quite clear that body size is a key factor, but it is unclear as to which factor is the most representative estimator for body size.

*Sirex nitobei* is a domestic woodwasp which is distributed in Japan (except Hokkaido Island) and China. *S. nitobei* have similar biology to *Sirex noctilio*. The life cycle of *S. nitobei* is usually completed in one year [14,15]. In Japan, *S. nitobei* may be responsible for the mortality of *Pinus densiflora* and *Pinus thunbergii* [16]. In China, *S. nitobei* attacks *Pinus tabuliformis* Carr., *Pinus sylvestris* var. *mongolica*., *Pinus armandii* Franch., *Pinus thunbergii* Parlatore., and *Pinus massoniana* Lamb. Substantial research has been performed on the oviposition activity and fecundity of *S. nitobei* [16,17,18]; however, there is insufficient research on the flight capacity of *S. nitobei*.

The main biological control agent of *S. noctilio*, *Deladenus siricidicola* Bedding (Tylenchida: Neotylenchidae), has two characteristic life cycles. In its mycetophagous life cycle, the nematode feeds on *Amylostereum areolatum* (Chaillet ex Fr.) Boidin (Russulales: Amylostereaceae) [17,18]. Under suitable conditions, the nematode transforms into an infective form and infects the *S. noctilio* larvae. *D. siricidicola* feed and reproduce within *S. noctilio* bodies, and eventually may sterilize *S. noctilio* by filling the eggs with nematodes [18,19]. Infestation by nematodes reduces the fecundity of *S. noctilio* [13,20] and also affects the individual size of *S. noctilio* [9,11]. A sterilizing strain of *D. siricidicola* has been found to parasitize *S. noctilio* in China. Another nematode, *Deladenus nitobei* Kanzaki, was found to infect *S. nitobei* in Takko, Aomori, Japan [21]. *S. nitobei* was infected with *D. siricidicola* Bedding and *D. nitobei* Kanzaki in Tongliao, Inner Mongolia Autonomous Region, China. The impact of *B. siricidicola* parasitism on woodwasp dispersal may alter the spatial transmission of the nematode, and as a result, affect the effectiveness of woodwasp pest control [9].

Less is known about the impact of nematode parasitism on the flight capacity of *S. nitobei*. Researchers also have different opinions on the impact of nematode infestation on the flight ability of *S. noctilio*. Bedding’s studies showed that nematode infestation does not affect the flight capacity of *S. noctilio* [8], while Villacide and Corley claimed that nematode infestation reduces its flight capacity [9]. Furthermore, Gaudon et al. argued that nematode-parasitized males fly farther than non-parasitized ones, However, there was no difference in the flight capacity between parasitized females and non-parasitized females [11]. In addition, there is variation in the research on the size of *S. noctilio* as influenced by nematode infestation. Parasitized *S. noctilio* were smaller than non-parasitized ones [9]. Other studies found that parasitized and non-parasitized females have a similar size, but parasitized males were larger than non-parasitized ones [13,22]. For *S. nitobei*, Kanzaki et al. reported that parasitized *S. nitobei* females were significantly smaller than non-parasitized ones, but parasitized males were larger than non-parasitized ones [23]. Over the past sixty years, tethered-flight mills have been developed to measure the flight capacity of insects in controlled settings [24]. The flight performance of insects on flight mills may vary from their natural flight behavior, due to artificial flight mechanics [24,25]. However, this method has several advantages over other methods of testing insect flight. Flight mills provide a more controlled and standardized environment, allowing researchers to examine the effect of factors on flight behavior, making it easier to obtain consistent results [26,27]. The substantial flight data of wasps were conducted by flight mills [9,10,11,13,28,29]. 

In the context of lacking flight data of *S. nitobei* and *S. noctilio*, and the diversity of opinions on the impact of nematode infestation on flight ability and body size, the objectives of this research were to: (1) use flight mills to measure and compare the flight capacity of *S. noctilio* and *S. nitobei*; (2) ascertain the effect of PED age on the flight capacity of *S. noctilio* and *S. nitobei*; (3) identify the impact of nematode infestation on the flight capacity of *S. noctilio* and *S. nitobei*; and (4) explore specific factors influencing the flight capacity of *S. noctilio* and *S. nitobei*.

## 2. Materials and Methods

### 2.1. Insect Rearing

*S. noctilio* adults were obtained from infested logs (*Pinus sylvestris* L. var. *mongolica* Litv.) collected from the Junde Forest Farm in Hegang, Heilongjiang Province, China (Lat: 47.16454383, Lon: 130.3633578). *S. nitobei* adults were obtained from infested logs (*Pinus tabuliformis* Carr.) collected from the Jinbaotun Forest Farm in Tongliao, Inner Mongolia Autonomous Region, China (Lat: 43.05027778, Lon: 123.4386111). All infested pine logs were cut into 1.5-meter-long pieces and then placed in an insect cage in the lab (temperature 25 °C, humidity 50–60%). Woodwasp adults were collected daily after emergence and reared alone in plastic jelly cups (aperture diameter 7 cm, base diameter 6 cm, height 3.9 cm) under the same conditions (temperature 25 °C, humidity 50–60%) as for the infested pine logs. 

### 2.2. Insect Measurement

Several morphological variables were measured to confirm the impact of morphological variation on flight performance. Body mass (BS) was measured using a digital analytical balance (0.0001 g) (BSA8201-CW, Sartorius Company). Pronotum width (PW), body length (BL), and forewing length (FL) were measured using a vernier caliper (0.01 cm) (PD-151, Prokit’s Industries Co. LTD., Shanghai, China). We tested both males and females, setting groups according to PED age. The *S. noctilio* and *S. nitobei* adults tested were 1~6 days post-eclosion. We defined three PED classes of adults, “Young” were 1~2 days post-eclosion adults. “Middle-aged” were 3~4 days post-eclosion adults, and “Old” were 5~6 days post-eclosion adults. In total, we used 98 males and 78 females of *S. noctilio* and 95 males and 75 females of *S. nitobei* in the flight tests. Among them, 26 females and 33 males of *S. noctilio* were “Young”, 26 females and 35 males of *S. noctilio* were “Middle-aged”, and 26 females and 30 males of *S. noctilio* were “Old”, while 26 females and 34 males of *S. nitobei* were “Young”, 24 females and 33 males of *S. nitobei* were “Middle-aged”, and 28 females and 35 males of *S. nitobei* were “Old”. All of the woodwasps were unmated and did not feed during the test. Each woodwasp was tested only once.

### 2.3. Flight Mill Recordings and Bio-Assay Procedures

The tethered-flight mill system (Jiaduo Industry and Trade Co., Ltd., Hebi, China) consisted of a flight mill, data collectors, environmental sensors, and supporting software. A 30-centimeter-long spinning stainless steel cantilever (diameter 1 mm) linked to a pivot was placed onto a base to provide near-frictionless flight. The flight mill system’s supporting software recorded the detailed flight parameters. 

Before the flight, the woodwasps were slightly anesthetized with 75% ether on the body surface, and we attached the rod to their pronotum quickly with cyanoacrylate glue. We attached plasticine on the other end of the rod to maintain balance. We stimulated flight by touching the tarsal of each woodwasp with a clean glass slide, and then started the recording software after the woodwasps initiated flight. All flight mills were located in a closed cubicle with controlled lighting, temperature, and humidity (200–350 Lux, 25 °C, and 50~60% RH). The flight recording of each woodwasp lasted for 12 h (9:00~21:00). After the flight, all woodwasps were frozen to death at −20 °C. We checked their abdomens for nematode infection by dissection. As the dissected woodwasps were recently dead, the presence or absence of nematodes could easily be confirmed by the detection of thread-like worms in the water solution with a stereoscopic microscope. 

The flight-mill software recorded four flight parameters (total flown distance, average single-flight distance, average single-flight duration, and mean speed). The total flown distance was the sum of the distances that the adult flew in 12 h. The average single-flight distance was the average distance of a single flight. The average single-flight duration was the average duration of a single flight. The mean speed was defined as the averaged speed across all flights during the test.

### 2.4. Statistical Analyses

Data analysis was performed using SPSS version 23.0 (IBM Corp., Armonk, NY, USA). An investigation of the effect of PED age (three cohorts) on total flown distance in both *S. noctilio* and *S. nitobei* was performed using the Kruskal–Wallis Test. The effect of parasitism status on flight performances was evaluated using the Mann–Whitney U test. The effect of body mass on flight performances was tested by the Spearman rank correlation test. A comparison of flight performance parameters between *S. noctilio* and *S. nitobei* was performed using the Mann–Whitney U test.

A generalized linear model (GLM) with a Gamma distribution and a log link function was employed to determine the effects of flight morphology traits on various flight performance parameters. We transformed the residual (RES) of ordinary least square method multiple regression (OLS) into 1/|RES|, which was used as a weight in weighted least square method multiple regression (WLS). Body length, pronotum width, forewing length, body mass, age cohorts, and parasitism status were put into multiple linear regression. We removed non-significant terms from the model in a stepwise manner until the maximal model remained. Then, the residuals of the models complied with the assumptions of normality and homoscedasticity. 

## 3. Results

### 3.1. Effect of the Post-Eclosion-Day Age on Flight Capacity

The PED age of woodwasps and the total flown distance were strongly correlated in *S. noctilio*, as observed by Spearman correlation (r = −0.616, *p* < 0.05). The Kruskal–Wallis H test showed that PED age significantly affected the total flown distance by *S. noctilio* females (H = 27.258, *p* < 0.001) and males (H = 19.776, *p* < 0.001) (Figure 1a, Table A1). As females and males of *S. noctilio* became older, the total flown distance declined gradually. The maximum total flown distance of *S. noctilio* females and males were 56.25 km and 30.80 km, respectively. The average single-flight distance (H = 18.186, *p* < 0.001), average single-flight duration (H = 15.633, *p* < 0.001), and mean speed (H = 14.670, *p* = 0.001) of *S. noctilio* females were significantly different for different PED ages (Figure 1b–d, Table A1). The average single-flight distance (H = 7.731, *p* = 0.021) and average single-flight duration (H = 7.375, *p* = 0.025) of *S. noctilio* males were also significantly different for different PED ages, while the mean speed (H = 3.401, *p* = 0.183) of *S. noctilio* males were not significantly different for different PED ages (Figure 1b–d, Table A1). 

No significant difference existed between different PED ages of *S. nitobei* for the total flown distance, average single-flight distance, average single-flight duration, or mean speed. The Kruskal–Wallis H test results are shown in Table A1.

### 3.2. Effect of Parasitism Status on Flight 

The parasitism status of *S. noctilio* females and males did not have a significant effect on the flight performance parameters of total flown distance, average single-flight distance, average single-flight duration, or mean speed (Table A2). 

The parasitism status of *S. nitobei* females and males did not have a significant effect on the flight performance parameters of total flown distance, average single-flight distance, average single-flight duration, or mean speed (Table A3). 

### 3.3. Effect of Body Mass on Flight

Linear correlation analysis indicated that the body mass of female *S. noctilio* was positively but weakly correlated with total flown distance (*r* = 0.3666, *p* = 0.001), and the body mass of male *S. noctilio* was positively moderately correlated with total flown distance (*r* = 0.4559, *p* = 0.0001). The body mass of male *S. noctilio* was positively moderately correlated with total flown distance (*r* = 0.4559, *p* = 0.001), average single-flight distance (*r* = 0.3276, *p* = 0.001), and mean speed (*r* = 0.3455, *p* = 0.0005). Linear correlation analysis indicated that the body mass of female *S. nitobei* was positively but weakly correlated with total flown distance (*r* = 0.3564, *p* = 0.0017), and positively moderately correlated with average single-flight distance (*r* = 0.4448, *p* = 0.0001), with average single-flight duration (*r* = 0.4036, *p* = 0.0003), and with mean speed (*r* = 0.4783, *p* = 0.0001). The body mass of male *S. noctilio* was positively but weakly correlated with total flown distance (*r* = 0.3297, *p* = 0.0011) and with average single-flight distance (*r* = 0.2045, *p* = 0.0468), and positively moderately correlated with average single-flight duration (*r* = 0.4448, *p* = 0.0001) and mean speed (*r* = 0.2868, *p* = 0.0048).

### 3.4. Comparison of the Flight Capacity of Sirex noctilio and Sirex nitobei

*S. noctilio* females and males in the field have a lifespan of 4 and 5 days, respectively [30]. Based on this study, *S. noctilio* and *S. nitobei* with an age of 1~2 PED had the maximum flight capabilities during their lifespans. It was meaningful to compare the flight capacity of *S. noctilio* and *S. nitobei* with an age of 1~2 PED.

Comparing the total flown distance, significant differences were observed between females of *S. noctilio* and *S. nitobei* (U = 161, z = −2.932, *p* = 0.003); the mean total flown distance of *S. noctilio* females (21.31 ± 3.28 km) was greater than that of *S. nitobei* females (14.13 ± 2.88 km). The average single-flight distance of *S. noctilio* females (1.58 ± 0.36 km) was greater than that of *S. nitobei* females (1.30 ± 0.64 km), with significant differences (U = 157, z = −3.01, *p* = 0.003). The average single-flight duration of *S. noctilio* females (0.47 ± 0.12 h) was greater than that of *S. nitobei* females (0.21 ± 0.06 h), with significant differences (U = 171.5, z = 2.729, *p* = 0.006). The mean speed of *S. noctilio* females (3.40 ± 0.24 km/h) was further than *S. nitobei* females (2.94 ± 0.24 km/h) with significant differences (U = 174, z = −2.68, *p* = 0.007) (Figure 2, Table A4).

The total flown distance of *S. noctilio* males (11.13 ± 1.37 km) was greater than *S. nitobei* males (5.68 ± 0.97 km) with significant differences (U = 305, z = −3.071, *p* = 0.002). Comparing the average single-flight distance, the average single-flight duration, and mean speed, no significant differences were observed between *S. noctilio* males and *S. nitobei* males (Figure 2, Table A4). Overall, the flight capacity of *S. noctilio* is greater than *S. nitobei* in both sexes. 

### 3.5. Main Factors Influencing Flight Capacity

Predictive explanatory models for the total flown distance of *S. noctilio* females were constructed. Age cohorts and body mass were used systematically to construct models, and other, non-significant, factors (nematode infestation, body length, pronotum width, forewing length) were excluded. The model was statistically significant and the model explanatory degree was good (R^2^ = 0.683, F = 80.668, *p* < 0.001). Among the variables, age cohort (Standardized β = −0.675) and body mass (Standardized β = 0.269) had the greatest effect on the total flown distance. Using the same statistical approach, in the predictive explanatory models for average single-flight distance, average single-flight duration, and mean speed of *S. noctilio* females, age cohorts had the greatest effect on average single-flight distance and mean speed (Table 1). In the predictive explanatory models for the total flown distance of *S. noctilio* males, body mass (Standardized β = 0.549) and age cohort (Standardized β = −0.507) were used systematically to construct models, and the model was statistically significant and the model explanatory degree was good (R^2^ = 0.737, F = 132.829, *p* < 0.001), indicating that body mass and age cohort had the greatest effect on the flown total distance. Using the same statistical approach, body mass and age cohorts had the greatest effect on average single-flight distance and average single-flight duration, while body mass had the greatest effect on mean speed (Table 1). 

For *S. nitobei*, only body mass was used systematically to construct predictive explanatory models for the total flown distance, average single-flight distance, average single-flight duration, and mean speed of both sexes, and the model was statistically significant and the model’s explanatory degree was acceptable (Table 1). 

## 4. Discussion

### 4.1. Flight Mill Techniques

Tracking insect flight is technically challenging, because of the small size of insects [31]. The methods for studying insect flight behavior include capture–mark–recapture [32,33], free-flight chambers [34,35], and tethered-flight systems [36,37]. On account of the lower recapture rates in the capture–mark–recapture method, it is not advisable to test invasive species such as *S. noctilio* in China [38]. For the free-flight chamber method, Gatehouse and Hackett argue that it is not suitable for fast and strong flying insects, considering the airflow control issues [39]. Tethered-flight systems are the most widely used tool for studying insect flight in the laboratory [24]. 

Although flight mills provide a convenient approach to measuring flight capacity, the output results may vary from those performed in nature. One significant difference is that insects on the flight mill do not have to generate the required lift they produce in free flight [25]. This was demonstrated in a study of locusts where the lift values of the tethered locusts were equivalent to approximately 70% of their body mass, which led to lower wing beat frequencies on flight mills than in free flight [39,40]. In another study, insects needed more energy to overcome the friction of the moving parts of the mill during flight [41]. This suggests that the flight speeds of insects measured on the flight mills may be less than those in free flight. Taylor et al. found that the flight speeds of *Agrilus planipennis* on the flight mills were approximately a third of the speeds performed in free flight [42].

In recent years, modified flight mills with lightweight arms on the magnetic suspension system have been designed to reduce the degree of mechanical friction [43]. Even if it is difficult to overcome some shortcomings of the flight mill system, it can provide a variety of measurements of the flight. These measurements (total flown distance, single-flight distance, single-flight duration, speed, and the number of flight bouts) are a reflection of flight capacity and migratory tendency. When conspecific insects fly on a flight mill, the flight distance and flight duration can be a relative indicator of natural flight behavior [24]. The flight mills make it convenient to classify migratory insects, and insects that perform long sustained flight indicate a possible migratory tendency. For example, the ‘one-hour rule’ is universally accepted to distinguish between migrants and non-migrants in *Melanoplus sanguinipes*. Individuals with more than 1 h single-flight duration in three trails were classified as migrants, whereas non-migrants usually fly less than 1 h [40,44]. In the gypsy moth, wing length was significantly correlated with their tethered-flight capacity, mainly in terms of total distance and maximum speed [27]. Moreover, wing length correlates with population density, which corresponds to population competition. Gypsy moths with relatively small wings are common in outbreaks and high-density populations [45,46].

In addition, flight mills provide an important device to evaluate insect dispersal performance offering the benchmarks of maximum inherent flight capacity [45]. Moreover, flight mills are operated under strictly controlled conditions, facilitating the detection of differences among endogenous factors (sex, age, individual size, etc.) and external factors (temperature, humidity, lighting, etc.) [46,47,48]. In this study, several endogenous factors were taken into account in the flight capacity of *S. noctilio* and *S. nitobei*, aiming to explore the factors affecting flight behavior. 

### 4.2. PED Age and Flight Capacity

In general, age significantly affects adult insects’ flight capacity [49]. In the present study, the flight capacity of *S. noctilio* females and males reduced with older age, with significant differences between different age cohorts. The newly emerged adults had a strong capacity to fly, owing to the fact that *S. noctilio* was sexually mature just after emergence [50]. However, *S. noctilio* did not feed after emergence, and they relied on their own fat energy reserves for locomotion and reproduction until death [51]. This was potentially the reason for their reduced flight capacity with increasing age. Meanwhile, PED age did not significantly affect the total flown distance, average single-flight distance, average single-flight duration, or mean speed for *S. nitobei*. Similarly, age did not affect flight capacity in male *Grapholitha molesta* [45]. 

### 4.3. Body Mass and the Flight Capacity

Several morphological traits contribute to the flight capacity, including the body size and loading of the wing [52,53]. The reproduction and dispersal abilities of insects are mostly influenced by body mass, as an adaptive factor that arises through complex hormonal interactions [54,55]. *S. noctilio* and *S. nitobei* did not feed after eclosion, so the body mass influenced their reproduction and activity [10,51]. In many studies of *S. noctilio*, individual size is an important factor that significantly affects flight capacity [9,10,11]. The parameters selected for individual size varied among studies, with some using body mass, and others using body length, pronotum width, etc. In the present study, four parameters representing individual size (body length, pronotum width, forewing length, and body mass) were measured and their correlation with and contribution to flight capacity were analyzed together. The results show that body mass was highly correlated in most flight capacity analyses. This effect is also because of the fact that body size is related to lipid stores, which power flight behavior. A higher body mass indicates more fat reserves and higher flight capacity. The body lipid content significantly influenced the distance flown of *Choristoneura conflictana* through interaction with sex and age [56]. Body mass affected the flown distance, and lipid content may also influence the flown distance. Moreover, body size could act as a limitation for maximum flight capacity [57]. As a result, body mass can serve as a proxy for individual size affecting the flight capacity of woodwasps.

### 4.4. Nematode Infestation and the Flight Capacity

Nematode infestation can affect individual size, either because nematodes and woodwasps compete with each other for nutrients or because nematodes limit the growth and development of woodwasps. In our study, nematode-infested *S. noctilio* females weighed significantly less than healthy females, and nematode-infested *S. nitobei* males were significantly heavier than healthy males. This result of *S. noctilio* is consistent with the result of Villacide and Corley’s study, and the reason for the different effects of nematode infestation on body mass of different species was unclear [9]. Villacide and Corley showed that nematode infestation reduces the flight capacity of *S. noctilio*, possibly due to the smaller size of nematode-infested *S. noctilio* [9]. Normally, nematode infestation does have a negative impact on woodwasps, such as body size and potential fecundity, but it may not have a significant negative effect on the flight capacity of woodwasps [11,13]. The observed decreasing in flight capacity is likely an indirect effect of nematode infestation, which may be varied with individual size and regional populations. Similar results have been shown in the monarch butterfly [9,58]. The inconsistency among earlier studies may be attributed to the different range of body size or genetic factors [11]. Based on the results of this study, nematode parasitism did not have a significant effect on the flight performance of *S. noctilio* and *S. nitobei*, which is consistent with the hypothesis of Gaudon et al. (parasitism infection has no adverse effect on the flight capacity of *S. noctilio*) [11]. Similar research reports that the nematode infection did not affect the total flight duration of the Douglas-fir beetle [59]. Akbulut and Linit found that a large proportion of *Monochamus carolinensis* adults carry a low nematode load and have unabated flight capacity, while a small proportion of *M. carolinensis* adults carry a high nematode load and have reduced flight capacity [60]. The nematode load of *S. noctilio* and *S. nitobei* comparative studies of the nematode load of woodwasp populations in different regions are needed in the future. Nematode-infested and healthy woodwasps were distributed in each concentration area, respectively. The limited influence of nematode infestation on woodwasp flight ensures that nematodes do not reduce the rate of spread of woodwasps and nematodes [61]. It is likely that the negative effect of nematodes on woodwasps was not conspicuous in the flight capacity, but research on the effects of nematode infestation on the dispersal of woodwasps should be continued in depth. Understanding the dispersal rate of *D. siricidicola* and the relationship between *D. siricidicola* and *S. noctilio* has had an important role in optimizing the use of nematodes to control *S. noctilio* [9,62]. 

### 4.5. Flight Capacity between Different Sexes

For both *S. noctilio* and *S. nitobei*, sex significantly affected the total flown distance, average single-flight distance, and average single-flight duration. The flight ability of females was generally higher than that of males. Sexual differences in intermittent flights may be related to reproduction, and a higher flight capacity helps insects to find the opposite sex to mate with and find suitable hosts to reproduce. Due to their naturally higher lipid stores, females are able to fly long distances [56]. Females may rely on carbohydrate energy rather than fat when flying long distances [63]. Furthermore, females may find it less energetically expensive to fly, due to their greater wingspan and potentially lower loading [64].

### 4.6. Flight Capacity between Two Sirex Species

In general, the flight capacity of *S. noctilio* surpassed that of *S. nitobei*. The flight capacity of female *S. noctilio* was significantly greater than that of *S. nitobei*. According to the comparison of the flight capacity of *S. noctilio* and *S. nitobei*, the dispersal of the *S. noctilio* population might be faster than that of *S. nitobei*. As a high-risk invasive pest, the dispersal speed of *S. noctilio* (57.0 km/year, 90% confidence interval 26.6 to 67.6 for the 95th quantile) was faster than thirteen exotic forest insects and diseases, such as pine shoot beetles, viburnum leaf beetles, and gypsy moths. [29]. 

### 4.7. Factors Affecting the Flight Capacity of Sirex noctilio and Sirex nitobei

Many studies have mentioned several endogenous factors such as age, nematode infestation, body mass, body length, pronotum width, forewing length, and metabolic rate, that may influence the flight capacity of woodwasps [9,10,11,13]. It is generally believed that wing shape is relevant to flight performance [65,66,67]. For butterflies, longer wing span was positively correlated with flight speed [68], the aspect ratio had a significant positive impact on acceleration capacity, and a more distant centroid was correlated with a stronger flight capacity [67]. The forewing length of woodwasps was taken into account in our study. However, no significant effect on the flight was found. Other measures of wings should be investigated in further research, such as forewing area, distance to the centroid, aspect ratio and wing loading, etc. 

According to the analysis of this study, PED age and body mass could explain a large part of the individual variations in flight, and they were important individual factors affecting the flight capacity of *S. noctilio*. This was partly consistent with Gaudon et al.’s research, which showed that body mass explains a portion of the flight capability of *S. noctilio* [11]. Larger individuals with larger wings fly faster, suggesting that more body fat reserves are acquired during larval development [28]. However, among the factors influencing the flight capacity of *S. nitobei*, body mass had the greatest effect, while factors such as PED age had little effect on flight capacity, probably because the metabolism of *S. noctilio* is faster and the effect of PED age is greater than that of *S. nitobei*. 

The present study focused on exploring the effects of individual factors on flight capacity, although flight capacity was also influenced by external factors, such as temperature and photoperiod. Lantschner et al. showed that temperature affects the dispersal of *S. noctilio*, and *S. noctilio* were more active and have higher dispersal rates in areas with higher annual temperatures [69]. Gaudon et al. reported an increase in *S. noctilio* flight speed and flight distance with increasing temperature [11]. The activity of *S. noctilio* is influenced by photoperiod—*S. noctilio* was active (fly at intervals) during the daytime and inactive (did not fly) at night under natural conditions [11]. The determination of factors related to the flight capacity of *S. noctilio* and *S. nitobei* can provide basic important data for predicting the potential establishment and spatial dynamics of woodwasps, and also provide ideas and suggestions for elucidating the mechanisms of breakout and monitoring.

## 5. Conclusions

In this study, a tethered-flight mill system was used to measure and compare the flight capacity of *S. noctilio* and *S. nitobei.* We specified that *S. noctilio* fly further and for longer than *S. nitobei*. We ascertained that PED age and body mass significantly influenced the flight capacity of *S. noctilio*, and their flight capacity decreased with PED age. While PED age did not significantly affect the flight capacity of *S. nitobei*, body mass significantly affected its flight capacity. We made it clear that parasitism infection did not significantly affect the flight performance of *S. noctilio* and *S. nitobei*. By means of multiple regression analysis, we confirmed that PED age and body mass were major factors affecting the flight capacity of *S. noctilio* and *S. nitobei*. These results provide substantial laboratory data on the flight studies of the two woodwasp species, which facilitates their risk analysis.

## Figures and Tables

**Figure 1 insects-14-00236-f001:**
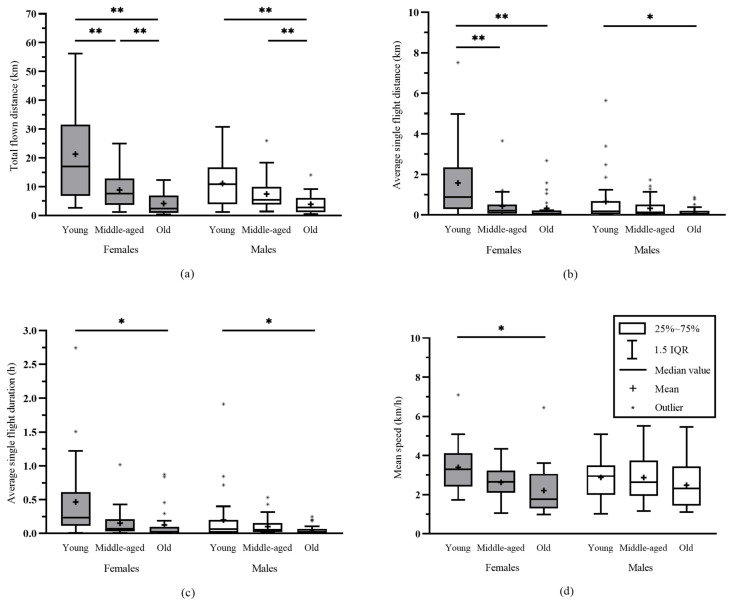
The flight capacity of *Sirex noctilio* for different PED ages. (**a**) Total flown distance of *S. noctilio* for different PED ages. (**b**) Average single-flight distance of *S. noctilio* for different PED ages. (**c**) Average single-flight duration of *S. noctilio* for different PED ages. (**d**) Mean speed of *S. noctilio* for different PED ages. * indicates a significant difference at *p* = 0.05; ** indicates a significant difference at *p* = 0.01.

**Figure 2 insects-14-00236-f002:**
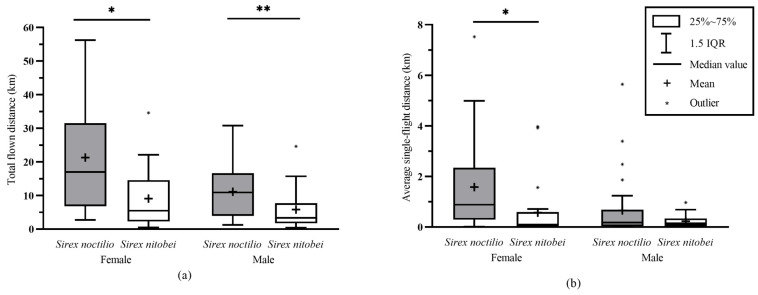
The comparison of the flight capacity of *Sirex noctilio* and *Sirex nitobei* (1–2 PED age). (**a**) The comparison of the total flown distance of *S. noctilio* and *S. nitobei*. (**b**) The comparison of the average single-flight distance of *S. noctilio* and *S. nitobei*. * indicates a significant difference at *p* = 0.05; ** indicates a significant difference at *p* = 0.01.

**Table 1 insects-14-00236-t001:** Interpretation model of flight performance parameters and independent variables in *Sirex noctilio* and *Sirex nitobei*.

	Dependent Variable	Model			Coefficients			
		R^2^	F	*p*	Independent Variables	Standardized β	*t*	*p*
*Sirex noctilio* females	Total flown distance	0.683	80.668	<0.001	Age cohorts	−0.675	−9.392	<0.001
					Body mass	0.269	3.748	<0.001
	Average single-flight distance	0.556	95.356	<0.001	Age cohorts	−0.746	−9.765	<0.001
	Average single-flight duration	0.594	111.39	<0.001	Age cohorts	−0.771	−10.554	<0.001
	Mean speed	0.49	73.01	<0.001	Age cohorts	−0.7	−8.545	<0.001
*Sirex noctilio* males	Total flown distance	0.737	132.829	<0.001	Body mass	0.549	9.878	<0.001
					Age cohorts	−0.507	−9.122	<0.001
	Average single-flight distance	0.715	119.382	<0.001	Body mass	0.552	7.601	<0.001
					Age cohorts	−0.374	−5.143	<0.001
	Average single-flight duration	0.457	40.017	<0.001	Body mass	0.474	5.799	<0.001
					Age cohorts	−0.333	−4.072	<0.001
	Mean speed	0.294	39.94	<0.001	Body mass	0.542	6.32	<0.001
*Sirex nitobei* females	Total flown distance	0.417	52.202	<0.001	Body mass	0.646	7.225	<0.001
	Average single-flight distance	0.4	48.677	<0.001	Body mass	0.632	6.977	<0.001
	Average single-flight duration	0.358	40.65	<0.001	Body mass	0.598	6.376	<0.001
	Mean speed	0.528	81.594	<0.001	Body mass	0.726	9.033	<0.001
*Sirex nitobei* males	Total flown distance	0.309	41.549	<0.001	Body mass	0.556	6.446	<0.001
	Average single-flight distance	0.378	56.565	<0.001	Body mass	0.615	7.521	<0.001
	Average single-flight duration	0.124	13.217	<0.001	Body mass	0.353	3.636	<0.001
	Mean speed	0.552	114.488	<0.001	Body mass	0.743	10.7	<0.001

## Data Availability

Not applicable.

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
