# Peer review of "Factors Affecting the Flight Capacity of Two Woodwasp Species, Sirex noctilio F. (Hymenoptera: Siricidae) and Sirex nitobei M. (Hymenoptera: Siricidae)"

_insects, 2023, doi:10.3390/insects14030236_

Round 1

Reviewer 1 Report (New Reviewer)

This manuscript describe comparative tethered flight performance in two species of woodwasp that are of agricultural importance.  Various features of flight speed and duration are usefully correlated with morphological features, and most interestingly with parasitism status, which was found to be non-significant relative to flight performance.  Sample sizes are sufficient, and the possible differences between tethered versus free flight measurements are recognized and discussed.  The paper in its current form is largely descriptive, and would benefit from more explicit hypothesis formulation (e.g., possible consequences of parasites), and from specific comparison with prior findings in this field; i.e., why are there conflicting earlier results on effects of parasitism, and why might there be no effect of parasites on tethered flight here (which is a somewhat counterintutive result)?  A broader literature survey of the effects of morphological variation on long-distance flight performance of other insect species would also make these specific results more relevant to our general understanding of insect flight; at present, the paper will mostly interest only woodwasp specialists.  Finally, there are a number of spelling errors (e.g., thethered rather than tethered at various places), and overall the manuscript would benefit from editing by a native English speaker. 

Author Response

Reviewer 2 Report (Previous Reviewer 2)

I have re-read the manuscript, and my mostly editorial comments have been addressed. The manuscript is acceptable from my point of view.

Author Response

We are grateful for your effort in reviewing our paper. Thank you for your positive comments. We have added some content to the discussion section (from Line345 to Line 373, and from Line 402 to Line 409.) and corrected a few language errors. We hope you are satisfied with our revised manuscript.

Reviewer 3 Report (Previous Reviewer 3)

This version does a good job of explaining the limitations of the method.  Other bits of the manuscript are also much improved, though I did find one typo.  On line 24, a "thethered-flight" should read a "tethered-flight".

Author Response

This manuscript is a resubmission of an earlier submission. The following is a list of the peer review reports and author responses from that submission.

Round 1

Reviewer 1 Report

The authors provided important information on flight ability of two woodwasps, Sirex noctilio and S. nitobei. Most of the data in this manuscript are convincing and presented by well-desined experiments. I did not observe any flaws.

Author Response

Dear reviewer:

We are grateful for your effort in reviewing our paper. Thank you for your positive comments. We have corrected a few language errors that we had missed in the original version. We hope you are satisfied with our revised manuscript.

Reviewer 2 Report

Liu et al focused on the flight behavior and capacity of Sirex noctilio and S. nitobei using flight mills. The authors found that S. noctilio was a stronger flyer than S. nitobei. Time after emergence affected the flight capacity of S. noctilio most, and the main factor affecting flight in S. nitobei was body mass. Overall, this is a good descriptive study and in consequence my comments are minor:   Abstract (suggestions for streamlining text): L21: causes pine mortality in plantations L23: flight capacity of the two wood wasps was studied L25: woodwasps were dissected to determine nematode infestation L25: post-eclosion-day (PED) L29: for both Sirex species   Introduction: L43: generally produces one generation per year L46: largely determined by L54: Write the genus name out when starting a new paragraph L55: similar biology as L56: may be responsible for L62: The main... L73: may alter the spatial transmission of the nematode L76: researchers also have... L79: reduces its flight capacity (delete of S. noctilio) L80: ones. However, there was no difference... L83: were smaller than L91: research were to: L94: specific factors influencing flight capacity   Materials and methods: L153: Why were data log transformed when all the statistical tests used were non-parametric?   Results: Can the authors rather place the labeling in Fig 1 and 2 (a, b, c, d) in the conventional left-hand corner of each panel in the figure? Section 3.2: This effectively is a list of negative results that is copied twice for the two insects. I would suggest just including one paragraph dealing with both insects at the same time (The parasitism status of S. noctilio and S. nitobei....), and for the statistical data refer to table A2, which should be moved to the main document.   Discussion: L276: I am not clear on which external factors the authors are referring to? Can you elaborate? L288-289: Could you substantiate the claim that S. nitobei uses less energy? It seems a bit far-fetched without any substantiation. Are the wasps smaller? L303: a higher body mass indicates more L307: body mass can serve as a proxy for L311: wood wasps compete with each other L312: wood wasps. Villacide showed L328: ...different sexes L345: delete etc. L352: delete JM

Reviewer 3 Report

Interesting work with some surprising results.  I enjoyed reading it.  

There are a number of small matters of language and punctuation to change.  For simplicity, I have listed the line number and written the corrected text.

Line: corrected text

38: delete comma after noctilio 

60: 18]; however, there is insufficient

105:  Woodwasp adults

109: 2.2 Insect measurement

113: a vernier

302: This effect is also because of the fact

323:. It is likely that

270: Tracking insect flight 

In the Materials and Methods section, the order of sentences implies the sequence of events.  Hence, it is unnecessary and distracting to place words like "Then" and "next" in the text.  So please correct the following.

132: and we attached

138: -20°C. We

Please be consistent in the use of units.  Either there is a space after the number or there isn't.  For example,

106: 7 cm

107: 3.9 cm

Line 247 begins a run-on sentence that needs to be broken into two or three smaller sentences for clarity.

Regarding line 128, 0.1 cm = 1 mm, so let's change the text to "diameter 1 mm".

When citing the research of others, attend to the number of authors.  For example, on line 352, " Gaudon JM’s research" should read "Gaudon et al.’s research", reflecting the multiple authors of the study.  All places where authors of studies are mentioned should be checked for consistency.  Typically, for a single-author paper, only the author's last name needs to be mentioned. However, last names of both authors need to be listed if there are two, and first author's last name  plus "et al." if more than two.

I was not certain that ether was a good choice for anesthesia, as I've only used CO2 and cold for this purpose in my work, but the fact that some individuals flew an incredible total of 60 km and as much as 5 km in a single flight mitigates against my caution.

I would rather see flight speed in units of m/s rather than km/h.  These are the properly reduced SI units. Let's not let the automobile industry guide our conventions of measurement.

I don't know why most of the data, Tables A1 - A4, are in an appendix.  As the substantive results of the study, they belong in the body of the paper.
